# A hybrid model- and deep learning-based framework for functional lung image synthesis from non-contrast multi-inflation CT

**Joshua R Astley**[1]                                    JASTLEY1@SHEFFIELD.AC.UK

[1]*POLARIS / Department of Oncology and Metabolism, The University of Sheffield, UK*

**Alberto M Biancardi**[1]                               A.BIANCARDI@SHEFFIELD.AC.UK

**Helen Marshall**[1]                                    H.MARSHALL@SHEFFIELD.AC.UK

**Guilhem J Collier**[1]                                 G.J.COLLIER@SHEFFIELD.AC.UK

**Paul JC Hughes**[1]                                    PAUL.HUGHES@SHEFFIELD.AC.UK

**Jim M Wild**[1]                                        J.M.WILD@SHEFFIELD.AC.UK

**Bilal A Tahir**[1]                                     B.TAHIR@SHEFFIELD.AC.UK

**Editors:** Under Review for MIDL 2021

## Abstract

Hyperpolarized gas MRI can visualize and quantify regional lung ventilation with exquisite detail but requires highly specialized equipment and exogenous contrast. Alternative, non-contrast techniques, including CT-based models of ventilation have shown moderate spatial correlations with hyperpolarized gas MRI. Here, we propose a hybrid framework that integrates CT-ventilation modelling and deep learning approaches. The hybrid model/DL framework generated synthetic ventilation images which accurately replicated gross ventilation defects in hyperpolarized gas MRI scans, significantly outperforming other model- and DL-only approaches. Our results show that a synergy between conventional CT-ventilation modelling and DL can improve the performance of functional lung image synthesis.

**Keywords:** Deep learning, CT ventilation, functional lung imaging, image synthesis.

## 1. Introduction

Hyperpolarized gas MRI can visualize and quantify regional lung ventilation with exquisite detail1, but requires highly specialized equipment and exogenous contrast. Alternative, non-contrast techniques, including multi-inflation CT-based models of ventilation, have shown moderate spatial correlation with hyperpolarized gas MRI (Tahir et al., 2018). Recent advances in deep learning (DL) using convolutional neural networks (CNNs) have shown promise for image synthesis applications, including the synthesis of functional lung images of single-photon emission computed tomography from CT (Liu et al., 2020). However, the synthesis of hyperpolarized gas MRI directly from CT has yet to be demonstrated. The relatively smaller datasets available for functional lung image synthesis mean that data-driven approaches alone are unlikely to generate realistic synthetic ventilation maps. We hypothesized that the combination of physiological modelling and data-driven DL approaches may leverage both methods' benefits to produce more physiologically plausible results. To this end, we propose a hybrid model/DL framework where conventional CT-based ventilation

models are used alongside structural inspiratory and expiratory CT scans as inputs to a deep CNN for functional lung image synthesis.

## 2. Methods

The dataset comprised paired inspiratory and expiratory CT and hyperpolarized helium-3 gas MRI ($^3$He MRI) for 47 patients with cystic fibrosis (n=19), asthma (n=12) or lung cancer (n=16). Inspiratory and expiratory CT scans were aligned using deformable image registration and subsequently registered to the spatial domain of $^3$He MRI via a corresponding anatomical proton MRI scan as previously described (Tahir et al., 2014). CT-based surrogate ventilation images were computed from the aligned inspiratory and expiratory CT images using a Hounsfield unit change ($CT^{HU}$) model-based metric (Tahir et al., 2018). We synthesized $^3$He MRI scans by training a 3D VNet CNN using TensorFlow 1.14. The CNN employs a root mean square error (MSE) loss with a PreLU activation and ADAM optimization on patches of 128x128x48 voxels with a batch size of two. The network was trained for 2150 epochs across 4 NVIDIA Tesla V100 GPUs. We evaluated four combinations of input channels for the CNN: 1) expiratory CT only; 2) inspiratory CT only; 3) expiratory CT and inspiratory CT; 4) inspiratory CT, expiratory CT and $CT^{HU}$ model. The correlations of the generated synthetic ventilation images from these four configurations and the $CT^{HU}$ model alone against corresponding $^3$He MRI scans were assessed at full resolution using Spearman's $\rho$ on all lung voxels. In addition, the DL models were evaluated with the voxel-wise MSE metric. 6-fold cross-validation was used and a paired t-test conducted across 47 patients to identify statistically significant differences between methods.

## 3. Results and Discussion

Figure 1 shows qualitative spatial agreement between $^3$He MRI and the hybrid model/DL method for three example cases, outperforming the CT-ventilation model and all other DL methods. Table 1 summarises descriptive statistics. The hybrid approach yielded statistically significant improvements in Spearman's $\rho$ compared to the $CT^{HU}$ model and all other DL methods ($p<0.0001$). Our results show that a hybrid model/DL framework, integrating CT-ventilation modelling as an input to a VNet CNN alongside multi-inflation CT, can yield physiologically plausible synthetic ventilation images that correlate well with corresponding hyperpolarized gas MRI scans.

Table 1: Descriptive statistics for the $CT^{HU}$ model and DL methods. The best $\rho$ and MSE values are shown in bold.

| Synthetic ventilation generation methods | Spearman's $\rho$ (Mean ± SD) | MSE (Mean ± SD) |
|---|---|---|
| $CT^{HU}$ model | 0.39 ± 0.18 | N/A |
| DL (inspiration CT) | 0.41 ± 0.18 | 0.032 ± 0.01 |
| DL (expiration CT) | 0.37 ± 0.20 | 0.027 ± 0.01 |
| DL (expiration CT + inspiration CT) | 0.42 ± 0.18 | 0.027 ± 0.01 |
| DL (expiration CT + inspiration CT + $CT^{HU}$ model | **0.46 ± 0.16** | **0.025 ± 0.01** |

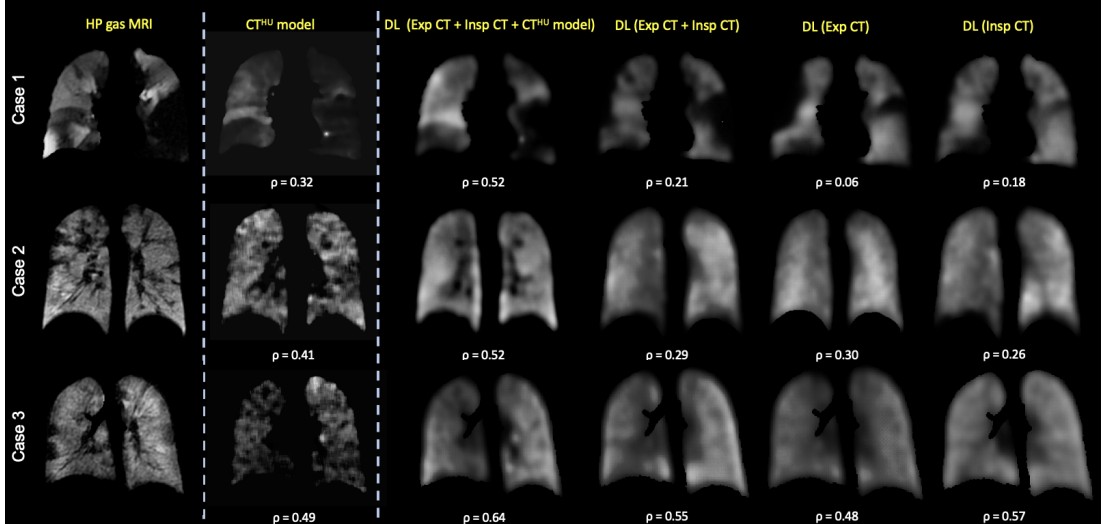

Figure 1: Example coronal slices from the $CT^{HU}$ model and four DL frameworks for three cases. Spearman's $\rho$ values between each method and hyperpolarized gas MRI are provided.

To the best of our knowledge, this study represents the first attempt to use deep learning to predict hyperpolarized gas MRI scans directly from CT. Our dataset contained images of patients with several respiratory diseases who were subject to a range of acquisition protocols, enhancing the proposed framework's robustness and generalizability. By using a combination of data-driven and model-based approaches, we make use of the relatively small datasets available for functional lung image synthesis. We demonstrate that leveraging the synergy between DL and CT ventilation modelling generates physiologically plausible synthetic ventilation scans across several diseases, indicating the potential of DL-based regional lung function from routinely acquired CT scans without exogenous contrast, which has implications for several clinical applications, such as functional lung avoidance radiotherapy.

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
