# OpenReview forum: "A hybrid model- and deep learning-based framework for functional lung image synthesis from non-contrast multi-inflation CT"
_MIDL.io/2021/Conference/Short — MIDL 2021 Poster_

### Official Review · Reviewer_QjTo · 2021-04-30

**Confidence:** 4
**Final Rating:** 3

**Summary:**

The authors present a 3D VNet based synthesis of He-MRI ventilation images based in-/ex-hale CT images.
Several input choices for the 3D VNet were compared (combinations of inspiration, exspiration images with and without Hounsfield difference based CT ventilation maps. Results are presented for a data set consiting of 47 patients with CT and He-MRI images available.

**Strengths:**

- Certainly an interesting approach
- CT based regional ventilation estimation is of interest for a variety of diseases
- Evaluation with Spearman's rank correlation makes sense
- results are somewhat promising

**Weaknesses:**

Unfortunately, the resuls of the DL method using only the CT-HU model is not presented. Why is that the case? The CT-HU model is the ventilation essence of the CT image pair and should in an ideal world correlate pretty much with true ventilation, as the He-MRI scan should do.
Clinical relevance is unclear and would need further assessment.

**Deanonymize Review:**

no

**Justification Of The Rating:**

The paper presents an academically interesting experiment asking if it is possible to synthesis an He-MRI ventilation image from CT image data (-> accept). The question however remains if it is also of clinical interest. To assess this, some clinical assessment would be necessary: The DL synthesized image has higher similarity to the He-MRI image than the CT-HU-difference image according to to the Spearman's correlation. But does it reveil additional clinical information which was not visible in the CT-HU image? (-> weak accept)



**Paper Type:**

validation/application paper

**Special Issue:**

no

---

### Meta-Review · Area_Chair_oyqo · 2021-05-10

**Recommendation:** Accept (Poster)
**Confidence:** 5

**Metareview:**

The meta-reviewer has studied the paper as well and supports the conclusion of the single reviewer. The paper is methodologically solid and presents an interesting approach, even if some questions are left open.

---

### Decision · Program_Chairs · 2021-05-11

Accept (Poster)